# Expression Dynamics Indicate Potential Roles of KIF17 for Nuclear Reshaping and Tail Formation during Spermiogenesis in *Phascolosoma esculenta*

**DOI:** 10.3390/ijms25010128

**Published:** 2023-12-21

**Authors:** Yue Pan, Jingqian Wang, Xinming Gao, Chen Du, Congcong Hou, Daojun Tang, Junquan Zhu

**Affiliations:** 1Key Laboratory of Aquacultural Biotechnology, Ministry of Education, Ningbo University, Ningbo 315211, China; cythiapam@163.com (Y.P.); wangjingqian0815@163.com (J.W.); gaoxinming@lsu.edu.cn (X.G.); 8788182@163.com (C.D.); houcongcong@nbu.edu.cn (C.H.); 2Key Laboratory of Marine Biotechnology of Zhejiang Province, College of Marine Sciences, Ningbo University, Ningbo 315211, China

**Keywords:** KIF17, spermiogenesis, spermatid remodeling, microtubule, *Phascolosoma esculenta*

## Abstract

Kinesin family member17 (KIF17), a homologous dimer of the kinesin-2 protein family, has important microtubule-dependent and -independent roles in spermiogenesis. Little is known about KIF17 in the mollusk, *Phascolosoma esculenta*, a newly developed mariculture species in China. Here, we cloned the open reading frame of *Pe-kif17* and its related gene, *Pe-act*, and performed bioinformatics analysis on both. *Pe*-KIF17 and *Pe*-ACT are structurally conserved, indicating that they may be functionally conserved. The expression pattern of *kif17/act* mRNA performed during spermiogenesis revealed their expression in diverse tissues, with the highest expression level in the coelomic fluid of *P. esculenta*. The expressions of *Pe-kif17* and *Pe-act* mRNA were relatively high during the breeding season (July–September), suggesting that *Pe*-KIF17/ACT may be involved in spermatogenesis, particularly during spermiogenesis. Further analysis of *Pe-kif17* mRNA via fluorescence in situ hybridization revealed the continuous expression of this mRNA during spermiogenesis, suggesting potential functions in this process. Immunofluorescence showed that *Pe*-KIF17 co-localized with α-tubulin and migrated from the perinuclear cytoplasm to one side of the spermatid, forming the sperm tail. *Pe*-KIF17 and *Pe*-ACT also colocalized. KIF17 may participate in spermiogenesis of *P. esculenta*, particularly in nuclear reshaping and tail formation by interacting with microtubule structures similar to the manchette. Moreover, *Pe*-KIF17 with *Pe*-ACT is also involved in nuclear reshaping and tail formation in the absence of microtubules. This study provides evidence for the role of KIF17 during spermiogenesis and provides theoretical data for studies of the reproductive biology of *P. esculenta*. These findings are important for spermatogenesis in mollusks.

## 1. Introduction

Spermiogenesis is the last stage of spermatogenesis and can yield mature and active sperm. Spermiogenesis involves three important cytological events: nuclear reshaping, acrosome formation, and tail formation [1]. During spermiogenesis, a structure termed the manchette forms transiently. The manchette consists of microtubules and actin filaments, in which the microtubule is formed by self-assembly of αβ microtubule heterodimers, combined with microtubule-associated proteins and motor proteins [2]. It has been suggested that the manchette is required as a transport track for spermatid head reshaping and tail formation [2,3]. Manchettes are involved in acrosome and tail formation, as well as nuclear reshaping during spermiogenesis [4,5,6]. In the azh mouse mutants, the abnormal location of the manchette in spermatids is consistent with a nuclear defect [7] and is associated with multiple coils in the sperm tail [2], indicating an indispensable role for the manchette in spermatid nucleus reshaping and tail formation.

The kinesin family of microtubule-dependent motor proteins can transport cargo along microtubules in eukaryotic cells using the energy generated by ATP hydrolysis [8]. Some kinesin proteins may be involved in nuclear reshaping, as well as acrosome and tail formation through the manchette structure during animal spermiogenesis [9,10]. For instance, in rats, kinesins may be involved in spermatid nucleus reshaping through the transient manchette [7,11,12]. Kinesin family member 17 (KIF17) is a member of the kinesin-2 family, which also includes KIF3A, KIF3B, and KIF3C [13]. KIF17 plays an important role in animal spermiogenesis [13]. The structural characteristics of KIF17 are consistent with those of other kinesin family members [14,15,16]. Interestingly, KIF17 plays a different role in spermiogenesis than traditional kinesin. KIF17 is involved in spermiogenesis via two mechanisms. In the first mechanism, KIF17 operates in a microtubule-dependent manner, similar to other kinesins [17,18]. In mice, KIF17 (designated KIF17b in the testes) co-localized with Spatial in the manchette structure, where the sperm nucleus is slightly elongated in shape than a sphere. As chromatin thickens, the manchette structure moves to one side of the nucleus and continues to be located in the principal piece of the sperm tail, suggesting that KIF17b may be involved in spermatid nucleus reshaping via the manchette [19]. In *Larimichthys polyactis* and *Larimichthys crocea*, KIF17 is involved in spermatid remodeling, including spermatid nucleus reshaping and tail formation through perinuclear microtubules during spermiogenesis [17,18]. Studies on vertebrates have shown that KIF17 and the manchette are involved in spermiogenesis; however, related studies in mollusks are yet to be conducted.

In the second mechanism, KIF17b is involved in spermiogenesis through a new mechanism independent of microtubule and kinesin motor domain regulation. KIF17b depends on the transcriptional regulation of cAMP-responsive element modulator (CREM)-activator CREM (ACT) (CREM-ACT) independent of microtubules [20,21,22,23]. Interestingly, in the testes, KIF17b interacts with ACT in spermatids after meiosis and affects CREM transcription in spermatids [23,24]. ACT (also known as four and a half LIM domains protein5, FHL5) belongs to the LIM-only protein family, whose LIM domain with double zinc-finger structures mediates interactions with other proteins [25,26]. Simultaneously, this special structure may make KIF17 different from the traditional transportation modes. The stalk of KIF17 binds to ACT, which can function as an ACT shuttle between the nucleus and cytoplasm, directly affecting the subcellular localization of ACT, thereby regulating CREM-dependent transcription in the testes [20,27]. This suggests that KIF17 and ACT may have a significant impact on the molecular mechanism of infertility. Both KIF17 and ACT play a vital role in spermiogenesis, and each are associated with a certain relationship. Thus, we suspect that KIF17 may play vital roles in both microtubule-dependent and -independent manners during spermiogenesis. However, this has not been reported in *P. esculenta.*

*P. esculenta* is an economically important aquatic organism on the southeast coast of China. Accordingly, research has increasingly focused on its developmental biology. According to a study on the ultrastructure of spermatids of *P. esculenta*, the spermatogenesis in *P. esculenta* particularly is complete in the gonads and coelomic fluid (CF) [28]. In addition, mitosis and differentiation of spermatogonia and meiosis of spermatocyte mainly occur in the testes, which are located at the base of the retractor muscle. Nevertheless, the process of spermatid development into mature sperm is completed in the CF. Spermatids are mainly located at the distal end of the mature testis and fall into the CF in the form of a spermatid mass, which develops into mature sperms in the CF and enters the nephridium for expulsion [28]. In addition, spermiogenesis in *P. esculenta* is accompanied by significant changes in microtubules and nuclear chromatin [29]. Currently, there are only a few studies on the mechanisms involved in spermatogenesis in *P. esculenta.* For example, KIFC1 was previously reported to be involved in nuclear reshaping and midpiece formation of spermatids in *P. esculenta* [29]. KIF3A/3B may play crucial roles in spermiogenesis, including acrosome biogenesis, sperm head reshaping, and enflagellation in *P. esculenta* [30]. However, the molecular mechanisms underlying the action of KIF17 during spermiogenesis in *P. esculenta* remain unclear.

In this study, the open reading frame (ORF) of *Pe*-*kif17/act* was cloned using specific primers. Bioinformatics analyses of the obtained sequences were performed. In addition, we analyzed the expression and distribution of *Pe*-*kif17/act* mRNA using semiquantitative RT-PCR, and *Pe*-*kif17* mRNA was detected using fluorescence in situ hybridization (FISH). Immunofluorescence (IF) labeling revealed that *Pe*-KIF17 colocalized with α-tubulin and *Pe*-ACT. Based on the distribution of microtubules with *Pe*-KIF17 and the co-localization of *Pe*-KIF17 with *Pe*-ACT during spermiogenesis, we hypothesized that KIF17 is involved in spermiogenesis in microtubule-dependent and -independent manner and established a functional model of *Pe*-KIF17. These studies provide basic data on the involvement of KIF17 in the reproductive biology of *P. esculenta*.

## 2. Results

### 2.1. Bioinformatic Analysis of the Pe-kif17 Gene

*Pe-kif17* is 2682 bp and encodes a protein of 894 aa with a molecular weight of 101.18 kDa and pI of 6.46 (Figure 1A). The predicted functional domain of the KIF17 protein was mainly composed of the N-terminal conserved motor domain (3–342 aa), the intermediate domain composed of two coiled-coil regions (392–444 aa/574–673 aa), and a C-terminal tail domain (673–894 aa) binding to cargo (Figure 1C). A predicted three-dimensional (3D) structural model of *Pe*-KIF17 was similar to the tertiary structure of KIF17 in other species (Figure 1D). We compared the homology of the predicted *Pe*-KIF17 aa sequence with that of other species (Figure 1B). The consensus and identity positions of the *Pe*-KIF17 aa sequence with its counterparts in *Homo sapiens*, *Mus musculus*, *Bos taurus*, *Danio rerio*, *Larimichthys crocea*, *Xenopus laevis*, *Bufo gargarizans*, *Mercenaria mercenaria*, *Crassostrea angulata*, and *Lingula anatina* were 52.6% and 41.7%, 53.4% and 42.4%, 52.2% and 40.4%, 59.5% and 46.4%, 57.9% and 45.7%, 59.7% and 49.3%, 62.0% and 50.6%, 60.3% and 50.7%, 54.8% and 45.3%, and 67.0% and 57.4%, respectively. The phylogenetic tree showed that *Pe*-KIF17 and *L. anatina* KIF17 were closely clustered and evolutionarily close, suggesting that their structures were conserved (Figure 1E).

### 2.2. Bioinformatic Analysis of Pe-act Gene

The ORF of *Pe*-act contains 1695 bp, coding for a 564 aa protein with a molecular weight of 64.45 kDa and pI of 5.85 (Figure 2A). The functional domain of the ACT protein is mainly composed of six LIM domains (Figure 2D). Each LIM domain with two zinc finger structures, as well as the 3D structural model of *Pe*-ACT, was constructed (Figure 2E), indicating that it was evolutionarily conserved. The homology of *Pe*-ACT amino acid sequence was compared with that of other species (Figure 2B). The similarity and consistency of *Pe*-ACT aa sequence with *H. sapiens*, *M. musculus*, *B. taurus*, *D. rerio*, *L. crocea*, *Salmo salar*, *Gallus gallus*, *Xenopus tropicalis*, *B. gargarizans*, and *Mytilus eduils* were 32.0% and 23.9%, 30.9% and 23.6%, 31.5% and 24.1%, 32.6% and 26.2%, 32.7% and 25.5%, 27.7% and 20.7%, 34.2% and 25.4%, 32.1% and 25.6%, 32.6% and 25.4%, and 65.5% and 53.9%, respectively. The phylogenetic tree showed that *Pe*-ACT was closely related to the *M. eduils* ACT, suggesting that its structure may be conserved (Figure 2C).

### 2.3. Expression Analysis of Pe-kif17 mRNA

We analyzed the relative expression levels of *Pe-kif17* mRNA in the CF, body wall, intestine, nephridium, and retractor muscles of *P. esculenta* using semiquantitative RT-PCR. *Pe-kif17* mRNA was expressed in all tissues. The expression level was highest in the CF compared with other tissues (Figure 3A). Similarly, we analyzed the relative expression of *Pe-kif17* mRNA in the CF at different stages of sexual gland development (March, May, July, August, September, and November) using semiquantitative RT-PCR. The results showed that *Pe-kif17* mRNA was expressed in the CF of *P. esculenta* in March, May, July, August, September, and November, with the highest expression observed in July (Figure 3B).

The spatiotemporal expression characteristics of *Pe-kif17* mRNA during spermiogenesis were detected by FISH (Figure 3C). In early spermatids, *Pe-kif17* mRNA were evenly distributed in the perinuclear cytoplasm (Figure 3C(A1–A4)). In middle spermatids, *Pe-kif17* mRNA signals were enhanced and concentrated around the nucleus (Figure 3C(B1–B4)). In late spermatids, *Pe-kif17* mRNA signals continued to be expressed around the nucleus (Figure 3C(C1–C4)). In mature sperm, *Pe-kif17* mRNA signals mainly converged to one side of the nucleus (Figure 3C(D1–D4)).

### 2.4. Expression Analysis of Pe-act mRNA

We analyzed the relative expression levels of *Pe-act* mRNA in the CF, body wall, intestine, nephridium, and retractor muscle of *P. esculenta* by semiquantitative RT-PCR. *Pe-act* mRNA was also expressed in all tissues (Figure 4A). The relative expression of *Pe-act* mRNA in the coelomic fluid in different stages of sexual gland development was determined (March, May, July, August, September, and November) by semiquantitative RT-PCR. *Pe-act* mRNA was expressed in the CF in March, May, July, August, September, and November (Figure 4B).

### 2.5. Preparation and Specificity of Pe-KIF17/ACT Antibodies

*Pe*-KIF17/ACT proteins were successfully expressed in *E. coli* Transetta (DE3) (Figure 5). SDS-PAGE clearly revealed bands with a molecular weight of 52.5 and 49.6 kDa in the induced cells, but not in the uninduced cells (Figure 5A,C). These two recombinant proteins were expressed in the supernatant and cell fractions. The purified recombinant proteins were each evident as a single band. Subsequently, the recombinant *Pe*-KIF17 and ACT protein was used to immunize rats and mice, respectively. Sera was collected to obtain rat polyclonal antibody against *Pe*-KIF17 and mouse polyclonal antibodies against *Pe*-ACT. Western blotting was performed to detect antibody specificity. The *Pe*-KIF17 antibody was detected as a single band of approximately 101.18kDa (Figure 5B). The *Pe*-ACT antibody was detected as two bands that matched the predicted molecular weight of the protein, at approximately 64.45 and 35–45 kDa (Figure 5D). These results suggest that rat anti-*Pe*-KIF17 and mouse anti-*Pe*-ACT polyclonal antibodies can be used for immunofluorescence assays.

### 2.6. Expression and Colocalization Characteristics of Pe-KIF17 with Tubulin during Spermiogenesis

The distribution of α-tubulin during spermiogenesis in *P. esculenta* (Figure 6) and co-localization characteristics of *Pe*-KIF17 with tubulin during spermiogenesis was observed by IF (Figure 7). In early spermatids, *Pe*-KIF17 and tubulin signals were evenly distributed in the cytoplasm (Figure 6A1,B1 and Figure 7A1–A5). In middle spermatids, *Pe*-KIF17 and tubulin signals colocalized similarly to early spermatids and clustered around the nucleus (Figure 6A2,B2 and Figure 7B1–B5). In late spermatids, *Pe*-KIF17 and tubulin signals migrated to one side of the nucleus, where the sperm tail formed (Figure 6A3,B3 and Figure 7C1–C5). In mature sperms, *Pe*-KIF17 and tubulin signals were detected in the tail (Figure 6A4,B4 and Figure 7D1–D5).

### 2.7. Colocalization between Pe-KIF17 and Pe-ACT in Spermiogenesis

Colocalization of *Pe*-KIF17 and *Pe*-ACT during spermiogenesis in *P. esculenta* was also observed by IF (Figure 8). In early spermatids, *Pe*-KIF17 and *Pe*-ACT signals were evenly distributed in the cytoplasm (Figure 8A1–A5). In middle spermatids, *Pe*-KIF17 and *Pe*-ACT signals were localized similarly to those in early spermatids and clustered around the nucleus (Figure 8B1–B5). In late spermatids, *Pe*-KIF17 and *Pe*-ACT signals migrated to one side of the nucleus where the sperm tail formed (Figure 8C1–C5). In mature sperm, *Pe*-KIF17 and *Pe*-ACT signals were detected in the tails of mature sperm (Figure 8D1–D5).

## 3. Discussion

### 3.1. Structural Features of KIF17 and ACT

KIF17 is a homologous dimer motor protein belonging to the kinesin-2 family. This protein consists of a microtubule-bound head motor domain, coiled-coil intermediate domain, and cargo-bound tail domain [14,15]. Its motor domain contains ATP hydrolysis sites and microtubule-binding sites, which can transport cargo to specific destinations in ATP hydrolysis-dependent manner [16]. Interestingly, the region where the KFI17b tail domain binds to the cargo does not bind to ACT. Its stem binds to ACT, enabling ACT transportation [27,31]. In ACT, the double zinc finger structure of the LIM domain is thought to mediate protein–protein interactions [25,26]. This particular interaction reflects the uniqueness of the KIF17b mediated transport of ACT [27]. In the present study, the predicted *Pe*-KIF17 protein had the same structural features as previously reported KIF17 [17,18]. Multiple sequence comparisons with KIF17 amino acid sequences from other species showed that their motor domains were highly conserved and contained microtubule-binding sites and ATP hydrolysis sites. These findings suggest that *Pe*-KIF17 might also use the energy generated by ATP hydrolysis to drive itself along microtubules through the motor domain. The structural conservation of *Pe*-KIF17 indicates that its function may have been evolutionarily conserved. Similarly, in this study, we found that the predicted *Pe*-ACT has six LIM domains, each of which has a structural motif consisting of two zinc fingers. The structurally conserved nature of both KIF17 and ACT suggests that the functions of both proteins may have been conserved during evolution.

### 3.2. Expression Characteristics of Pe-kif17/act mRNA

KIF17 is expressed at high levels in the gray matter of the brain, where it possibly plays a crucial role for neurons [14,31]. In mice, KIF17 is highly expressed in the brain and is involved in neuronal events required for learning and memory via the transport of the basic N-methyl-D-aspartate-type glutamate receptor (NR2B) [32,33]. In addition, KIF17 is highly expressed in the testes [20]. Saade et al. [19] reported that KIF17b and Spatial function in spermatid reshaping in mouse testes. This coincides with the high expression of *kif17* mRNA in stage IV of *L. polyactis* and *L. crocea* spermiogenesis. The testes of both contain numerous spermatids [17,18,34], which suggests the involvement in spermiogenesis, due to the high expression of KIF17 in the testes of *L. crocea* and *L. polyactis* [17,18]. Corresponding findings regarding the expression of *kif17* mRNA revealed the highest expression in the CF, where spermatids underwent morphological transformation into sperm throughout spermiogenesis, in contrast to other tissues, during July of the breeding season. Interestingly, spermatid masses, blood cells, and granular cells were also observed in the CF. Moreover, the proportions of these three cell types vary between the months. In the reproductive season (July–September), spermatid masses are the main components of the CF, whereas in the nonreproductive season (March, May, and November), spermatid masses are relatively small [28,35,36]. In addition, we studied the expression and distribution characteristics of *kif17* mRNA during spermiogenesis. *Pe*-*kif17* mRNA signals were continuously detected during spermiogenesis. The expression pattern of KIF17 provides an important basis for our research on KIF17 in *P. esculenta*.

ACT is expressed in the germ cells of male mice and may play an important role in sperm function [37]. Using FISH, we detected a strong *act* mRNA signal in human, mouse, and cynomolgus monkey round sperm cells. The findings indicate that *act* plays an important role in spermiogenesis [22]. In this study, semiquantitative RT-PCR analysis revealed that *Pe*-*act* mRNA was distributed in all tissues and was expressed in the CF in different months. These findings provided basic data for us to explore the association between KIF17 and ACT. Subsequently, we used IF to determine the colocalization of both.

### 3.3. Pe-KIF17 May Be Involved in Nuclear Reshaping and Tail Formation in a Microtubule-Dependent Manner

The manchette plays an important role in the reshaping of the spermatid nucleus and is a type of cytoskeleton composed of microtubules [2,5]. In mice, Kotaja et al. [27] found that KIF17 signals were colocalized with α-tubulin, suggesting that KIF17 was related to the manchette. In the present study, we observed the distribution of α-tubulin during spermiogenesis in *P. esculenta* (Figure 6). Tubulin signals were randomly distributed in the perinuclear cytoplasm of the early spermatids (Figure 6A1,B1). Subsequently, tubulin signals were closely associated with signal intensification in the middle spermatids (Figure 6A2,B2). However, the tubulin signals gradually weakened in the final spermatids (Figure 6A3,B3). Ultimately, tubulin signals were mainly concentrated in the tail of mature sperm, suggesting that there may be microtubule structures similar to those of manchettes in spermatids (Figure 6A4,B4).

Moreover, we also observed that *Pe*-KIF17 co-localized with α-tubulin in the process of spermatid nucleus reshaping, and the protein signals gradually transferred from the perinuclear cytoplasm to one side of the sperm tail, with obvious changes in spermatid nucleus. These findings suggest that *Pe*-KIF17 may interact with microtubule structures similar to manchette to participate in spermatid remodeling (Figure 7). Previous studies have found that KIF17b and Spatial also exist in the manchette region and are continuously expressed in the sperm tails [19] and that KIF17b is expressed throughout spermatogenesis persists in the sperm tail [20,38]. Furthermore, in *L. polyactis* and *L. crocea*, KIF17 is involved in spermatid remodeling, including spermatid nucleus reshaping and sperm tail formation, through perinuclear microtubules during spermiogenesis [17,18]. Based on these findings, a model involving *Pe*-KIF17 in spermatid remodeling can be proposed (Figure 9A1,A2). Thus, it can be speculated that KIF17 may be involved in spermatid remodeling through a microtubule structure similar to the manchette.

### 3.4. Pe-KIF17 and Pe-ACT May Participate in Nuclear Reshaping and Tail Formation Independently of Microtubules

ACT (FHL5) belongs to the LIM-only protein family. The LIM domain, composed of approximately 50 amino acid residues, is rich in cysteine, forming two zinc finger structures. Its conserved cysteine and histidine residues bind to Zn^2+^ to form a stable tertiary protein structure, which is the main region mediating interactions with other proteins [25,26]. Studies in male mice have found that the sperm cell transcription mechanism is the transcriptional activation of CREM by binding to the activator ACT, without requiring phosphorylation at serine 117 as a transcriptional activator or association with CBP. These findings demonstrate the crucial role of ACT in male reproduction [21,22,23,24]. Moreover, in ACT^−/−^ mice, a major abnormal phenomenon is the presence of a large number of tails with hairpin rings and sperm head deformity, indicating that ACT may play an important role in sperm tail formation and spermatid nucleus reshaping [38]. Studies in male mice have found that KIF17b, an isoform of KIF17, can also participate in spermatogenesis independent of microtubules and domains, by determining the subcellular localization of ACT (activator of CREM in testis) and participating in the transcription of related genes in spermiogenesis [21,22,23,27]. These findings indicate that KIF17b with ACT plays an important role in spermiogenesis.

We assessed the specificity of the *Pe*-KIF17 and ACT antibodies. Previous studies reported that antibody against KIF17 could also recognize KIF17b [20,27]. In the present study, the antibody specificity of *Pe*-ACT in this study was consistent with previous research [20,38], indicating the value of the *Pe*-KIF17 and *Pe*-ACT antibodies we generated for IF. These antibodies could be used for IF, which revealed signals of KIF17 and ACT that were randomly distributed in the perinuclear cytoplasm throughout the process from early spermiogenesis to mature sperm. The signals initially strengthened, gradually weakened, and finally became concentrated in the tail of mature sperm, especially in the middle piece of the sperm tail. These observations are similar to those of previous IF studies. In mice, KIF17b and ACT display coupled intracellular localization in male germ cells, where ACT is located in the nuclei of round spermatids, with KIF17b present in both the nucleus and cytoplasm [20]. In round spermatids, when CREM-dependent transcription stops and the nuclei of sperm begin to elongate, ACT migrates to the cytoplasm along with KIF17b and persists in the sperm tail [20,27,38]. Therefore, KIF17 and ACT may be involved in the formation of the sperm tail. Based on the above information, *Pe*-KIF17/ACT may be involved in sperm nuclear reshaping and tail formation during spermiogenesis in *P. esculenta*. A model involving *Pe*-KIF17 in spermatid remodeling is shown in Figure 9B1, B2. This model needs to be confirmed by techniques that include gene knockout and RNA interference.

## 4. Materials and Methods

### 4.1. Tissue Sampling

*P. esculenta* individuals weighing 2.9 to 4.8 g were obtained from Xiangshan (Zhejiang, China). The CF of male *P. esculenta* were obtained in March, May, July, August, September, and November. In addition to CF with mainly spermatid mass, body wall, intestine, retractor muscle, and nephridium were used as experimental materials in July of the reproductive season [35] for RNA and protein extraction. Simultaneously, a portion of the CF was handled in phosphate-buffered saline (PBS) containing 4% paraformaldehyde (PFA), and frozen sections were prepared for FISH and IF. Mature sperm were obtained by artificial induction [28], and the CF after successful induction treatment is described above.

### 4.2. RNA and Protein Extraction 

Total RNA and protein were extracted from the CF, body wall, intestine, nephridium and retractor muscle using the TRIzol reagent (Invitrogen, Carlsbad, CA, USA) and RIPA lysis buffer (Beyotime, Shanghai, China) according to the manufactures’ instructions. The PrimeScript RT kit (TaKaRa Bio, Dalian, China) was used for reverse transcription. The complimentary DNA (cDNA) reverse transcription product was stored at −20 °C.

### 4.3. Cloning of kif17/act (ORF) 

The intermediate segment sequence of *kif17/act* was obtained from the transcriptome sequence of *P. esculenta*. According to this sequence, specific primers were designed by Primer 5.0 software (Table 1). Touchdown PCR was performed using 94 °C for 5 min, 8 cycles of 94 °C for 30 s, 56 °C (each cycle reduced by 0.5 °C) and 72 °C for 1min; 27 cycles of 94 °C for 30 s, 53 °C for 30 s and 72 °C for 1 min; and a final extension at 72 °C for 10 min. Electrophoresis was performed on 1.0% agarose gel to resolve the DNA fragments, which were then extracted, purified, and ligated to the vector, and used to transform *Escherichia. coli* Trans-5α. The bacteria with the target band were selected and sent to Youkang Biology (Hangzhou, China) for sequencing.

### 4.4. Bioinformatics Analysis of KIF17/ACT 

*Pe*-KIF17/ACT sequences were manipulated using the Sequence Manipulation Suite (http://www.bio-soft.net/sms/index, accessed on 21 April 2023). Multiple sequence alignments of *Pe*-KIF17/ACT amino acid sequences were performed using Vector NTI 10. Phylogenetic trees for KIF17/ACT were constructed using the adjacency method in MEGA 5.1. The ProtParam tool was used to calculate the molecular weights and isoelectric points (https://web.expasy.org/protparam/, accessed on 21 April 2023). Using the NCBI tool (https://www.ncbi.nlm.nih.gov/Structure/cdd/wrpsb.cgi, accessed on 21 April 2023) and online tools (https://zhanglab.ccmb.med.umich.edu/I-TASSER, accessed on 21 April 2023), we predicted the KIF17/ACT functional domain and tertiary structure of the KIF17/ACT proteins.

### 4.5. Semiquantitative RT-PCR Analysis of kif17/act mRNA Expression 

The relative expression of *kif17/act* mRNA in the CF from different months (March, May, July, August, September, and November) and different tissues (CF, body wall, intestine, nephridium, and retractor muscle) was determined using semiquantitative RT-PCR. *Pe*-KIF17/ACT specific primers were designed using Primer 5 (Table 1) and were synthesized by Youkang Biology. The glyceraldehyde 3-phosphate dehydrogenase (*gapdh*) gene was used as the positive control. The relative expression of *Pe-kif17* and *Pe-act* mRNAs was analyzed using Image J. The SPSS version 25.0 software (NIH, Bethesda, MD, USA) was used to determine the differences in expression levels of *kif17/act* mRNA in the CF of different months and different tissues of *P. esculenta* using means of one-way analysis of variance and Tukey’s multiple comparison. *p* < 0.05 indicated a significant difference.

### 4.6. FISH

As previously described by Zhao et al. [39], spermatids from the CF of *P. esculenta* were extracted and embedded in O.C.T compound. The samples were first stored at −20 °C for 1–2 h and then stored at −80 °C. The prepared samples were cut into 5 um thin slices using a freezing microtome (Thermo Fisher Scientific, Waltham, MA, USA). The sections were then attached to adhesion microscope sections and stored at −80 °C. The KIF17 probe (Table 1) was synthesized by GENEWIZ (Suzhou, China) and combined with fluorescein isothiocyanate (FITC). Its specificity was examined using the BLAST function of NCBI. The sections were examined by confocal laser scanning microscopy (CLSM) using the LSM880 microscope (Carl Zeiss, Jena, Germany) and imaged.

### 4.7. Antibodies

*Pe*-KIF17/ACT recombinant proteins were prepared and purified using a prokaryotic expression technique for animal immunization to obtain anti-*Pe*-KIF17 and anti-*Pe*-ACT antibodies. First, the full-length sequence of *Pe*-KIF17/ACT ORF (894aa) and (564aa) were obtained. The polypeptides were approximately 101.18 kDa and 64.45 kDa, respectively. The primers were as follows: anti-KIF17-F, GGCAACACAAAGACGCTCA; anti-KIF17-R, TTCTCCACCCACCATTTGT; anti-ACT-F, CCAGACGAAGATGCGGATAC; and anti-ACT-R, GAAAATCAACTCATCACAGGCT. The sequence was ligated to the pET-28a-SUMO vector and expressed in receptive *E. coli* Trans-5α. After PCR identification using T7 F/R primers, the bacterial suspension was sent to Youkang Biosciences for sequencing. The plasmid was extracted from the *E. coli* Trans-5α bacterial suspension carrying the correct sequence. The extracted plasmids were then introduced into Transetta (DE3) receptor cells (DE3-KIF17/ACT) for proliferation and induced by 0.1 M isopropyl β-D-1-thiogalactopyranoside (IPTG). Recombinant proteins were extracted and purified using a His-tag protein purification kit (Soluble Protein/Inclusion Body Protein; Beyotime, Shanghai, China) and identified using polyacrylamide gel electrophoresis. Finally, rats and mice were immunized with *Pe*-KIF17/ACT recombinant protein. Western blotting and immunofluorescence of extracted sera were performed as previously described [40]. In addition, we purchased commercial antibodies (all from Beyotime, Shanghai, China), which included rabbit anti α-tubulin antibody (Cat. No. AG0126), horseradish (HRP)-conjugated goat anti-rat IgG (H + L) (Cat. No. A0192), HRP-conjugated goat anti-mouse IgG (H + L) (Cat. No. A0216), Cy3-conjugated goat anti-rat IgG (H + L) (Cat. No. A0507), FITC-conjugated goat anti-mouse IgG (H + L) (Cat. No. A0568), and Alexa Fluor488 conjugated goat anti-rabbit IgG (H + L) (Cat. No. A0423).

### 4.8. Western Blotting

The specificity of *Pe*-KIF17/ACT antibodies was detected by Western blotting. First, total protein was extracted from the CF of *P. esculenta* using RIPA lysis buffer (Solarbio, Shanghai, China) and PMSF solution (Beyotime, Shanghai, China), and the protein concentration was determined using the BCA method (Kangweishiji, Jiangsu, China). The protein was diluted with 5× Sodium dodecyl sulfate-polyacrylamide gel electrophoresis (SDS-PAGE) (Beyotime, Shanghai, China) and boiled for 10 min for structural denaturation. The proteins were separated isolated on a 4–20% gel (ACE, Nanjing, China) and transferred to PVDF membranes (Bio-Rad, Hercules, CA, USA). After 2 h blocking with 5% skim milk solution at room temperature, the membranes were incubated with rat anti-*Pe*-KIF17 and mice anti-*Pe*-ACT antibodies at 4 °C overnight, and the antibodies were diluted at 1:200 by 0.1% Tris buffered saline containing Tween (TBST) solution. The membranes were washed four times with 0.1% TBST for 15 min each time. The membranes were then incubated with HRP-conjugated goat anti-rat IgG (H + L) and HRP-conjugated goat anti-mouse IgG (H + L) antibodies diluted 1:1500 in 0.1% TBST for 2 h at 37 °C. The membranes were washed four times with 0.1% TBST for 15 min each time. Finally, the membranes were imaged using a chemiluminescence imager (Tanon 5200, Shanghai, China).

### 4.9. IF

As previously described [41], 4% PFA-PBS was used to embed the CF of *P. esculenta* at 4 °C overnight. The fixed sample was transferred to 0.5 M sucrose in PBS (pH 7.4) at 4 °C for 2 h. Finally, the sample was embedded in O.C.T. compound, which was used for frozen sections as described in Section 4.6 and stored at −80 °C until required. At that time, the frozen sections were dried at room temperature for 10 min, permeabilized with 0.3% Tween in PBS (PBST) for 12 min and blocked with 5% bovine serum albumin (BSA) at room temperature for 2 h. The blocked sections were incubated with rat anti-*Pe*-KIF17 antibody diluted 1: 70 in 3% BSA) and rabbit anti α-tubulin antibodies (diluted at 1:100 by 3% BSA at 4 °C overnight followed by washing four times with 0.1% PBST for 10 min each time. The treated sections were incubated for 1 h with Cy3-conjugated goat anti-rat IgG (H + L) and Alexa Fluor488 conjugated goat anti-rabbit IgG (H + L) (both diluted 1: 500 in 3% BSA) and washed four times with 0.1% PBST for 10 min each time. Finally, the treated sections were incubated with 4′,6-diamidino-2-phenylindole (DAPI; Beyotime, Shanghai, China) for 7 min, mounted with Antifade Mounting Medium (Beyotime, Shanghai, China), and imaged by CLSM using the aforementioned LSM880 microscope.

In addition, as previously described [42], frozen sections retrieved from −80 °C storage were dried at room temperature for 10 min, permeated with 0.3% PBST for 12 min, then blocked with 5% BSA in 0.1% PBST for 2 h at 37 °C. The sections were then incubated with rat anti-*Pe*-KIF17 antibody diluted 1:70 using 3% BSA overnight at 4 °C. The sections were rewarmed at room temperature for 30 min then washed four times with PBS. The sections were then incubated with Cy3-conjugated goat anti-rat IgG (H + L) diluted 1:100 using 3% BSA at 37 °C for 1.5 h in a dark environment and washed five times with PBS. The treated sections were again blocked in the presence of 5% BSA at 37 °C for 2 h. These treated slices were incubated with mouse anti-ACT antibody diluted 1:70 with 3% BSA at 37 °C for 2 h and washed five times with PBS, incubated with FITC-conjugated goat anti-mouse IgG (H + L) diluted 1:100 with 3% BSA at 37 °C for 2 h, and washed five times with PBS. After incubation with DAPI (Beyotime, Shanghai, China) for 7 min, the treated sections were mounted with Antifade Mounting Medium (Beyotime, Shanghai, China) imaged by CLSM using the LSM880 microscope.

## 5. Conclusions

The ORFs of *Pe-kif17* and *act* were cloned. The expression patterns and potential function of the *Pe*-KIF17 and ACT proteins during spermiogenesis were analyzed. *Pe-kif17* and *Pe-act* mRNA were maximally expressed in the CF, with a large number of spermatids in the reproductive season, whose characteristics were perhaps essential for spermiogenesis. In addition, IF revealed the colocalization of *Pe*-KIF17 and tubulin and that the signal migrated from the perinuclear cytoplasm to the side of the sperm tail during spermiogenesis. Moreover, *Pe*-KIF17 with *Pe*-ACT colocalized during spermiogenesis. These results suggest that KIF17 may be involved in spermatid remodeling, including sperm nucleus reshaping and tail formation, either through interactions with a microtubule structure similar to that of the manchette or through its interaction with ACT. The detailed function of KIF17 in the spermiogenesis process of *P. esculenta* requires further study.

## Figures and Tables

**Figure 1 ijms-25-00128-f001:**
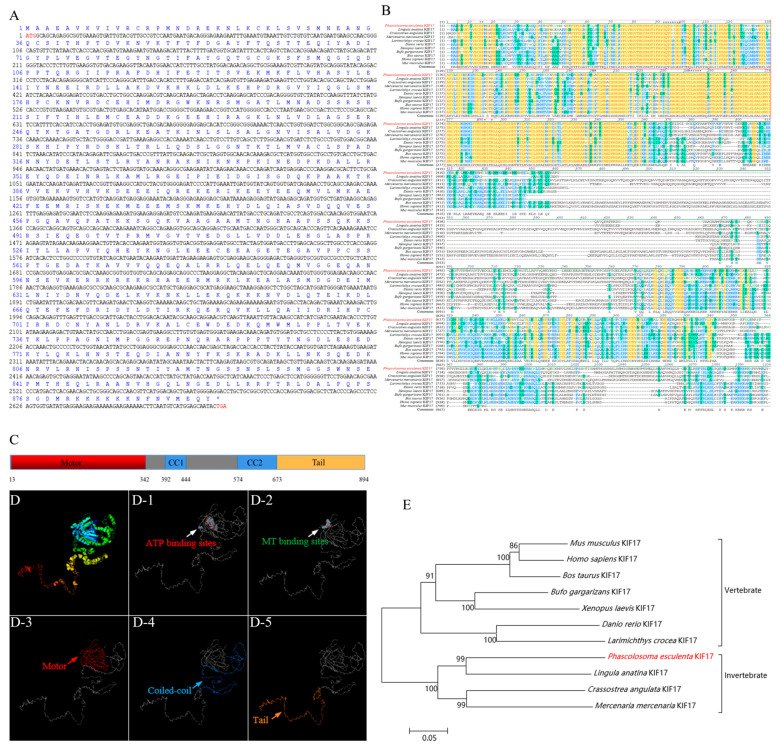
Analysis of *Pe*-*kif17* ORF characteristics. (**A**) ORF of *Pe-kif17* with the deduced amino acid sequence, and it contains 2682 bp and can encode 894 amino acids. The initiation codon is ATG and the termination codon is TGA (both marked in red). (**B**) Multiple sequence alignment of KIF17 amino acid sequence of *P. esculenta* and other species. The identical amino acids are represented in yellow regions; more than 50% similarity is represented in blue regions, and a lower similarity is represented in green regions. The motor domain is marked in a red box. Red triangles indicate microtubule-binding sites. Blue triangles indicate putative ATP hydrolysis sites. (**C**) Protein structure domain of *Pe*-KIF17. (**D**) Predicted tertiary structure of *Pe*-KIF17; (**D-1**) predicted ATP hydrolysis sites; (**D-2**) predicted microtubule-binding sites; (**D-3**) predicted N-terminal motor domain (red); (**D-4**) predicted intermediate domain (blue); and (**D-5**) predicted C-terminal tail domain (orange). Motor: N-terminal head motor domain; CC1: coiled-coil region 1; CC2: coiled-coil region 2; Tail: C-terminal tail domain. (**E**) Phylogenetic tree of *Pe*-KIF17 homologous proteins.

**Figure 2 ijms-25-00128-f002:**
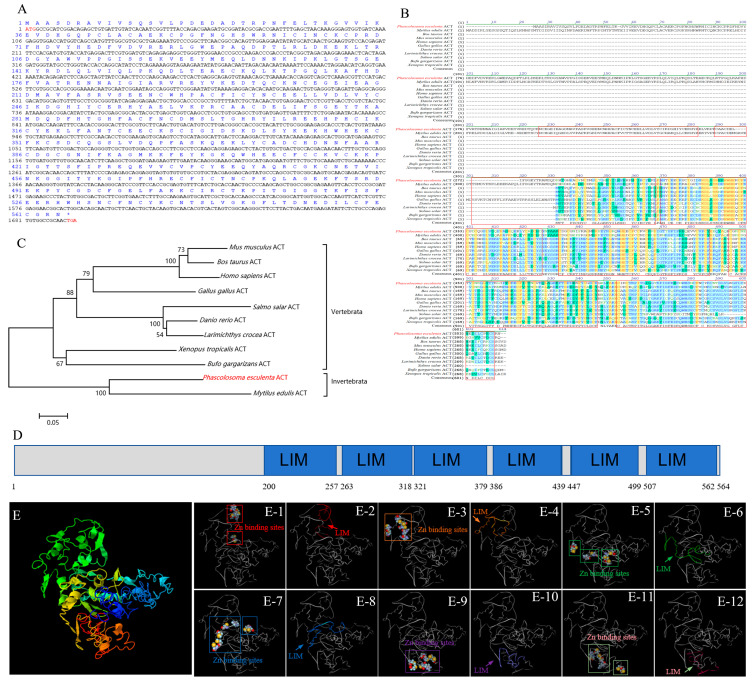
Analysis of *Pe*-*act* ORF characteristics. (**A**) ORF of *Pe-kif17* of 1695 bp encoding 564 amino acids. The initiation codon is ATG and the termination codon is TGA (both marked in red). (**B**) Multiple sequence alignment of ACT amino acid sequence of *P. esculenta* and other species. The LIM domain is marked in a red box. (**C**) Phylogenetic tree of *Pe*-KIF17 homologous proteins. (**D**) Protein structure domain of *Pe*-ACT with six LIM domains. (**E**) Predicted tertiary structure of *Pe*-ACT; (**E-1**,**E-3**,**E-5**,**E-7**,**E-9**,**E-11**) predicted the double zinc finger structure; (**E-2**,**E-4**,**E-6**,**E-8**,**E-10**,**E-12**) predicted the LIM domain.

**Figure 3 ijms-25-00128-f003:**
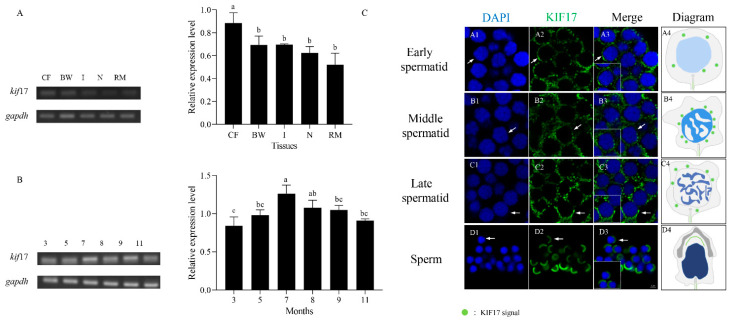
The relative expression level of *Pe*-*kif17* mRNA. (**A**) The relative expression of *Pe*-*kif17* mRNA in different tissues of *P. esculenta*, the highest expression level of *kif17* mRNA was found in CF. Abbreviations: CF: coelomic fluid; BW: body wall; I: intestine; N: nephridium; and RM: retractor muscle. (**B**) The relative expression of *Pe*-*kif17* mRNA in the CF of *P. esculenta* in different months. The highest expression of *kif17* mRNA was found in July. Values are expressed as mean ± SD, with lower-case letters indicating significant differences between groups (*p* < 0.05). (**C**) Fluorescence in situ hybridization results of *Pe-kif17* mRNA during spermiogenesis in *P. esculenta*. (**A1**–**A4**) In the early spermatid, *Pe-kif17* mRNA signals were distributed in the perinuclear cytoplasm. (**B1**–**B4**) In the middle spermatid, *Pe-kif17* mRNA signals were enhanced and distributed around the nucleus. (**C1**–**C4**) In the late spermatid, *Pe-kif17* mRNA signals continued to express around the nucleus. (**D1**–**D4**) In the mature sperm, the signals were mainly concentrated on one side of the nucleus. Blue signal represents nuclei stained with 4′, 6-diamidino-2-phenylindole (DAPI). *Pe-kif17* mRNA was stained with FITC labeled probe (green). The scale bar is 2 µm.

**Figure 4 ijms-25-00128-f004:**
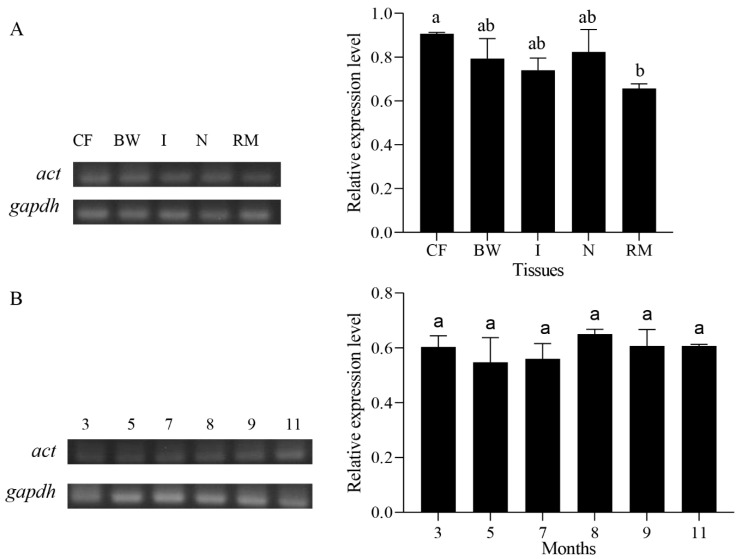
The relative expression level of *Pe*-*act* mRNA. (**A**) The relative expression of *Pe*-*act* mRNA in different tissues of *P. esculenta*. The highest expression level of *act* mRNA was found in the CF. Abbreviations: CF: coelomic fluid; BW: body wall; I: intestine; N: nephridium; and RM: retractor muscle. (**B**) The relative expression of *Pe*-*act* mRNA in the CF of *P. esculenta* in different months. Values were expressed as mean ± SD. Lower case letters indicate significant differences between groups (*p* < 0.05).

**Figure 5 ijms-25-00128-f005:**
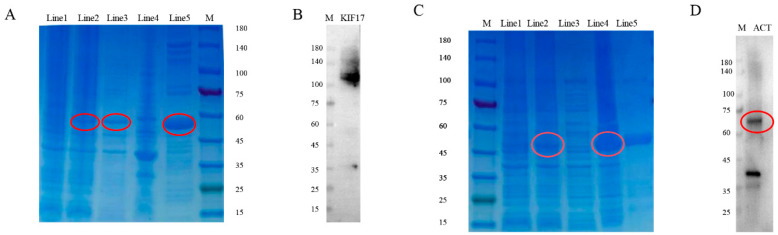
Expression and purification of *Pe*-KIF17 and ACT. (**A**) Lane 1: uninduced total protein; Lane 2: total protein expression of KIF17 induced by IPTG; Lane 3: soluble protein; Lane 4: inclusion body protein; Lane 5: purified recombinant protein with molecular weight of 52.53 kDa; and Lane 6: protein markers. (**B**) Specificity of anti-*Pe*-KIF17 antibody in rats as detected by Western blotting. Only a single band with a molecular weight of approximately 101.18 kDa is detected, which is consistent with the predicted molecular weight of *Pe*-KIF17, confirming the specificity of the antibody. (**C**) Lane 1: protein marker; Lane 2: uninduced total protein; Lane 3: total protein expression of ACT induced by IPTG; Lane 4: soluble protein; Lane 5: inclusion body protein; Line 6: purified recombinant protein with molecular weight of 49.6 kDa. (**D**) Specificity of anti-*Pe*-ACT antibody in mice as detected by Western blotting. A single band with a molecular weight of about 64.45 kDa is detected, which is consistent with the predicted molecular weight of *Pe*-ACT.

**Figure 6 ijms-25-00128-f006:**
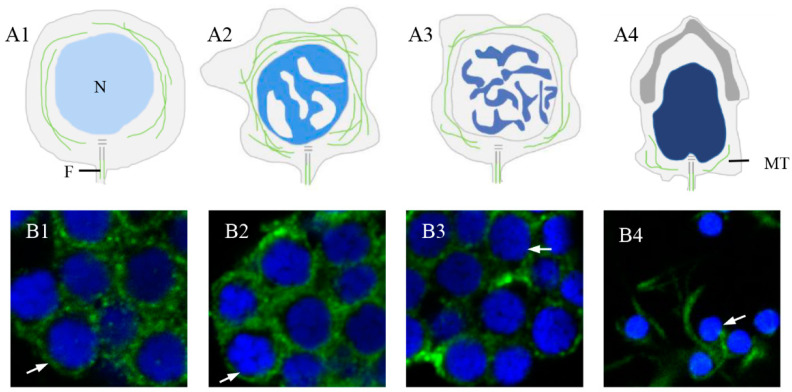
Spermiogenesis in *P. esculenta*. (**A1**,**B1**) early spermatid; (**A2**,**B2**) middle spermatid; (**A3**,**B3**) late spermatid; (**A4**,**B4**) mature sperm. Abbreviations: N: nucleus; F: flagellum; MT: microtubule.

**Figure 7 ijms-25-00128-f007:**
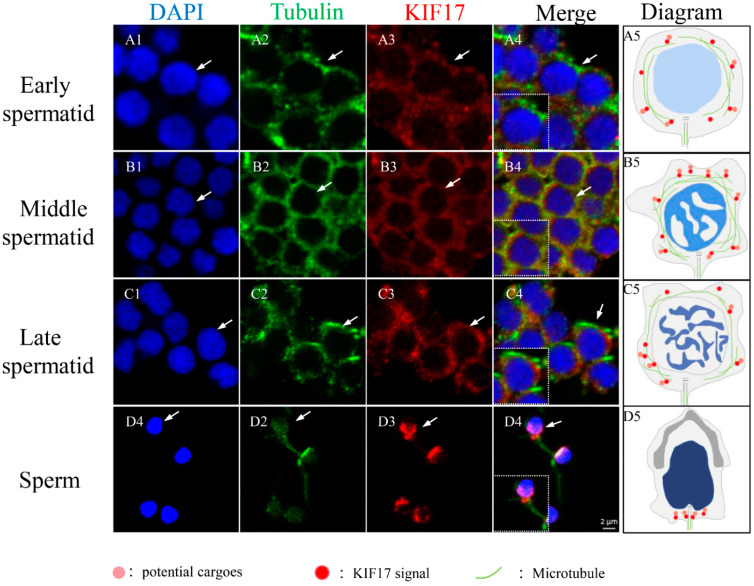
Immunofluorescence localization of *Pe*-KIF17 with tubulin during spermiogenesis in *P. esculenta*. (**A1**–**A5**) In early spermatid, *Pe*-KIF17 protein signals were randomly distributed in the perinuclear cytoplasm. (**B1**–**B5**) In middle spermatid, *Pe*-KIF17 protein signals were enhanced and distributed around the nucleus. (**C1**–**C5**) In late spermatid, *Pe*-KIF17 protein signals moved to one side of the nucleus where sperm tail formed. (**D1**–**D5**) In mature sperm, the signal was mainly located in the sperm tail. Nuclei were stained with DAPI (blue). *Pe*-KIF17 was stained with Alexa Fluor Cy3 conjugated antibody (red). Tubulin was stained with AlexaFluor488 conjugated antibody (green). The scale bar is 2 µm.

**Figure 8 ijms-25-00128-f008:**
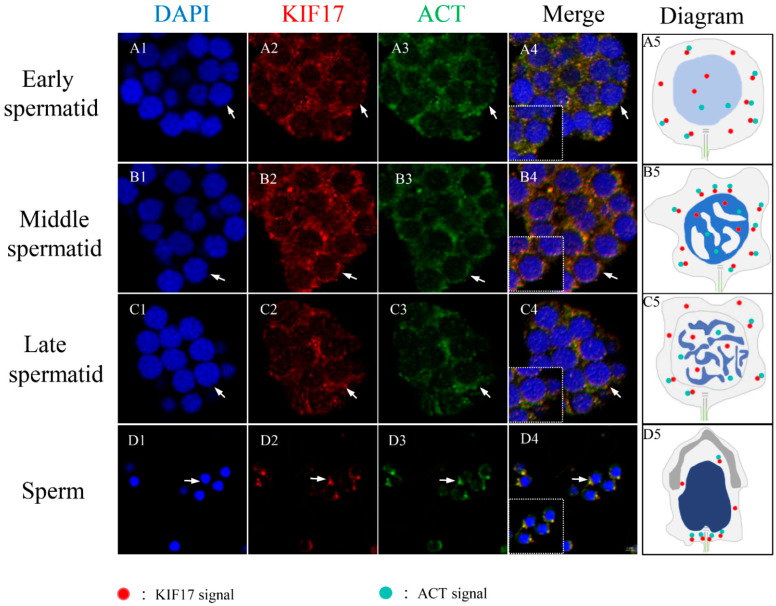
Immunofluorescence localization of *Pe*-KIF17 with *Pe*-ACT during spermiogenesis in *P. esculenta*. (**A1**–**A5**) In early spermatid, *Pe*-KIF17 and *Pe*-ACT signals were randomly distributed in the perinuclear cytoplasm. (**B1**–**B5**) In middle spermatid. *Pe*-KIF17 and *Pe*-ACT signals were localized similarly to those in early spermatids and clustered around the nucleus. (**C1**–**C5**) In late spermatid, *Pe*-KIF17 and *Pe*-ACT signals moved to one side of the nucleus where sperm tail formed. (**D1**–**D5**) In mature sperm, the signal was mainly located in the sperm tail. Nuclei were stained with DAPI (blue). *Pe*-KIF17 was stained with Alexa Fluor Cy3 conjugated antibody (red). *Pe*-ACT was stained with Alexa Fluor FITC conjugated antibody (green). The scale bar is 2 µm.

**Figure 9 ijms-25-00128-f009:**
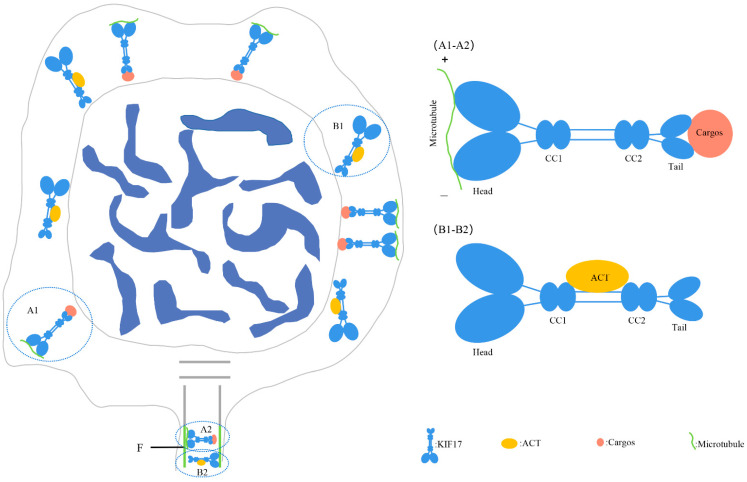
(**A1**,**A2**) KIF17 may be involved in spermatid reshaping, including nuclear reshaping and tail formation, by interacting with microtubule structures similar to the manchette. (**B1,B2**) KIF17 with ACT may be involved in nuclear reshaping and tail formation independently of microtubules. Head: N-terminal head motor domain; CC1: coiled-coil region 1; CC2: coiled-coil region 2; Tail: C-terminal tail domain; F: flagellum.

**Table 1 ijms-25-00128-t001:** The sequences of primers and probes.

Primers/Probe	Sequences (5′-3′)	Purpose
KIF17-F1	GAATGGTAGGTGTGACGG	PCR
KIF17-F2	AGGAACAAATGGTGGGTG	PCR
KIF17-R1	TGGTCATAGCATAGATGGTGT	PCR
KIF17-R2	TTGCTCTGTGCTGTTGTG	PCR
KIF17-RT-F	AGGCCCTAAGGAGGCTACAA	Semiquantitative RT-PCR
KIF17-RT-R	CCCATCCATGCTAGCCAGAG	Semiquantitative RT-PCR
ACT-F1	CGGGAAGAGCACCCTCACTGTA	PCR
ACT-F2	CAAGGCTGGGATGAAGAAGTTT	PCR
ACT-R1	CGGGATGCCCTTGTAGGTGATA	PCR
ACT-R2	TCCTCGAACGAGATGAACTTGG	PCR
ACT-RT-F	GGCTGGGATGAAGAAGTTT	Semiquantitative RT-PCR
ACT-RT-R	GTTGTGCCGATGGGTTT	Semiquantitative RT-PCR
GAPDH-F	CTGGTGAAGTTGGAGAAAAAG	Semiquantitative RT-PCR
GAPDH-R	GCTGAAGGAGCAGAGATGAT	Semiquantitative RT-PCR
KIF17 Probe sequence	GCTATATCTTGCTCTGTGCTGTTGTGTAGTTT	Fluorescence in situ hybridization

## Data Availability

The authors confirm that the data supporting the findings of this study are available within the article.

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
