# Peer review of "Expression Dynamics Indicate Potential Roles of KIF17 for Nuclear Reshaping and Tail Formation during Spermiogenesis in *Phascolosoma esculenta"

_ijms, 2023, doi:10.3390/ijms25010128_

Round 1
Reviewer 1 Report
Comments and Suggestions for Authors
In this manuscript the authors study KIF17, a homologous dimer of the kinesin-2 protein family in the mollusk, Phascolosoma esculenta, notably in spermiogenesis. This last part is novel in mollusks, and the structural data is interesting (Fig 1-3). While there is some interesting data (although the paper is not properly organized in terms of Figures and their order) I do not think the manuscript reaches the potential for IJMS for several reasons,
-the main issue is that the authors speculate on function without having any functional data, but merely correlations.
-in some cases the differences discussed are either small (FIg 4, 5A) or even not existent (Fig 5B), and it is not clear what the specific relevance of KIF17 may be.
-Fig 4 is about mRNA localization, not protein. Yet it is discussed as if it were functionally relevant.
-Protein localization images by immunofluorescence (Fig 7,8) are particularly poor. This is worrysome given that confocal microscopy was used. Better images must be provided.
-Both Introduction and Discussion are very long and speculative given the type of data the authors have.
Comments on the Quality of English LanguageThe text should be fully revised for many sentence construction issues. automatic translators may have been used.
Author Response
The answer to Q1: In this study, only gene expression was achieved, because KIF17 is relatively conserved in structure, based on comparison with homologous proteins from other species. It is assumed that its function is also relatively conserved. Due to the difficulty of spermatids culture of P. esculenta, the culture has not been successful so far, and it is still difficult to explore the function of KIF17. If the cell line is established, the related functions of KIF17 can be verified, and this study will provide basic data for the future research of KIF17 in reproductive biology.
The answer to Q2: Thank you very much for your advice. First of all, the pictures have been reordered. Figure 3A shows the expression of kif17 mRNA in various tissues of P. esculenta. There are obvious differences between CF (coelomic fluid) and other tissues, p<0.05. Meanwhile, CF is the site of spermatids development. kif17 mRNA was highly expressed in July of the reproductive season as shown in Figure 3B (spermatids were the main component in the coelomic fluid in July), indicating that KIF17 may be related to spermiogenesis, and we subsequently performed KIF17 protein localization (IF). Figure 4 is about the expression of ACT (KIF17 related gene), which shows the expression of act mRNA in different tissues of P. esculenta (Figure 4A) and the expression of act mRNA in coelomic fluid of P. esculenta in different months (Figure 4B), with no significant difference. The description of the two results in the previous paper is not accurate enough, and I have revised the results in the paper. The results show that act mRNA is expressed in various tissues of P. esculenta and in the coelomic fluid of P. esculenta in different months. The reason why we will do the ACT protein localization experiment later is that studies have shown that KIF17b carries ACT between the nucleus and the cytoplasm, which directly affects the subcellular localization of ACT and regulates the transcription dependent on CREM (ACT is the activator of CREM) in testis, so the expression of ACT in tissues is meaningful.
The answer to Q3: Your question is reasonable. I have removed too many speculative conclusions from the paper.
The answer to Q4: I'm sorry that these pictures are not clear enough. I have reprocessed the picture 7 and picture 8 and partially enlarged the picture.
The answer to Q5: According to your suggestions, I have deleted a lot of speculative content from the introduction and discussion, so as to make the content of the paper more concise and rigorous.

Reviewer 2 Report
Comments and Suggestions for Authors
The authors should be congratulated for their work and for addressing an important topic. Only a few points warrant mention:
Minor comments:
1. In the “Introduction” section, “With the KIF3A of the spermatid from mice knocked out and the manchette abnormal, it results in deformed spermatid heads, which means that KIF3A may mediate manchette structure to participate in the spermatid nucleus reshaping”, the phrase is not clear. The machette is abnormal because of KIF3A or is another abnormality, independent from this KIF3A ko? I suggest to clarify it in the text.
2. In the “Introduction” section, the introduction on KIF17 is redundant, a revision of the style is required. Similarly, the introduction of the rationale behind the study on p.esculenta should be revised in its form.
Comments on the Quality of English LanguageEnglish editing is required.
Author Response
The answer to Q1: I'm very sorry for my incorrect statement. My expression was not clear enough, which led to your misunderstanding. I would like to express that the knockout of KIF3A in mice causes serious damage to the sperm tail formation, also affects the composition of the manchette structure and the formation of the sperm head. I incorporated this idea into the viewpoint that the kinesin family might be involved in the spermatids reshaping.
The answer to Q2: According to your suggestions, I have deleted a lot of speculative content from the introduction and discussion to make the paper more concise and rigorous.

Reviewer 3 Report
Comments and Suggestions for Authors
The manuscript presents a compelling study on the structural features, expression patterns, and potential functions of Pe-KIF17 and Pe-ACT during spermiogenesis in P. esculenta. The authors have provided a comprehensive analysis of the molecular interactions and evolutionary conservation of these proteins, which is commendable. The use of immunofluorescence to elucidate the localization and interaction of these proteins during spermatid remodeling is particularly noteworthy and adds significant value to the understanding of spermiogenesis.
Specific Comments:
Clarity and Structure:
The manuscript is well-structured, with clear subdivisions that guide the reader through the study's rationale, methods, results, and conclusions. However, the authors might consider providing a more detailed background on the significance of KIF17 and ACT in the broader context of spermatogenesis to enhance the manuscript's accessibility to a wider audience.
Methodological Rigor:
The methods employed, particularly the immunofluorescence techniques, are well-chosen for the study's objectives. It would be beneficial if the authors could include more detailed information on the controls used in their experiments to strengthen the validity of their findings.
Interpretation of Results:
The discussion provides a thoughtful interpretation of the results, linking the structural features of KIF17 and ACT to their potential functions. However, the authors should consider discussing alternative hypotheses or potential limitations of their study to provide a balanced view.
Evolutionary Perspective:
The manuscript touches upon the evolutionary conservation of the proteins in question. Expanding on this aspect by including a phylogenetic analysis could offer deeper insights into the evolutionary significance of the findings.
Future Directions:
While the conclusions drawn are supported by the data, the manuscript would benefit from a section that explicitly outlines future research directions, such as potential functional assays or genetic studies to validate the roles of KIF17 and ACT in spermiogenesis.
Technical Specificity:
The specificity of the antibodies used for Pe-KIF17 and Pe-ACT should be addressed more thoroughly. The authors mention the antibodies' specificity but do not provide detailed evidence to support this claim. Including such validation data would reinforce the credibility of the immunofluorescence findings.
Suggestions for Improvement:
The authors might consider expanding their discussion on the implications of their findings for the field of reproductive biology and potential applications in addressing fertility issues.
A more detailed examination of the interaction between KIF17 and ACT, potentially through co-immunoprecipitation or other biochemical assays, could provide stronger evidence for their cooperative role in spermiogenesis.
Conclusion:
This manuscript makes a valuable contribution to the field of molecular biology, particularly in understanding the molecular mechanisms underlying spermiogenesis. With the incorporation of the suggested revisions and additional data, the manuscript would be strengthened significantly, making it a solid candidate for publication.
Comments on the Quality of English LanguageThe English language quality appears to be generally good, with appropriate use of technical terminology and a formal academic tone.
Author Response
The answer to Q1: Thank you very much for your suggestions. I have carefully revised the discussion to strengthen relevance, deleted some speculative views, and discussed the impact on the field of reproductive biology.
The answer to Q2: Thank you very much for your comments. I have done co-immunoprecipitation of KIF17 and ACT many times before, but no result of interaction between the two was detected. We analyzed the possible reasons as follows: 1. KIF17 and ACT bind to each other instantaneously, resulting in weak affinity of interacting proteins, making it difficult to observe results. 2. The reason for the small amount of combination of KIF17 and ACT makes it difficult to observe the results of interaction between the two. This also illustrates the difficulty of verifying in vivo, and we are also working on the establishment of cell lines, but without success. If the cell line can be successfully established, we will conduct co-ip and pull-down experiments again and publish them in the next paper specializing in functional studies.

Round 2
Reviewer 1 Report
Comments and Suggestions for Authors
The authors have unfortunately not adequately addressed my previous concerns.
Round 3
Reviewer 1 Report
Comments and Suggestions for Authors
Unfortunately my opinion has not changed.
